# Convergent and environmentally associated chromatic polymorphism in *Bryconops* Kner, 1858 (Ostariophysi: Characiformes: Iguanodectidae)

Andressa S. Gonçalves[1], André L. Netto-Ferreira[2], Samantha C. Saldanha[1], Ana C. G. Rocha[1], Suellen M. Gales[1], Derlan J. F. Silva[1], Daniel C. Carvalho[3], João B. L. Sales[1], Tibério C. T. Burlamaqui[1,4]*, Jonathan S. Ready[1]

1 Group for Integrated Biological Investigation (GIBI), Center for Advanced Biodiversity Studies (CEABIO), Biological Sciences Institute, Federal University of Pará (UFPA), Belém, Pará, Brazil, 2 Laboratory of Ichthyology, Zoology Department, Biological Sciences Institute, Federal University of Rio Grande do Sul (UFRGS), Porto Alegre, Rio Grande do Sul, Brazil, 3 Laboratório de Genética da Conservação, Programa de Pós Graduação em Biologia dos Vertebrados, Pontifícia Universidade Católica de Minas Gerais, Belo Horizonte, Minas Gerais, Brazil, 4 Instituto Tecnologico Vale, Belém, Pará, Brazil

* tburla@gmail.com

**Data Availability Statement:** All relevant data are within the paper and its Supporting Information files.

## Abstract

*Bryconops* Kner, 1858, includes two well defined subgenera based on morphological evidence, with each containing at least one species (*B. (Bryconops) caudomaculatus* and *B. (Creatochanes) melanurus*) with a very wide distribution, within which regional populations present color variations. To test if phenotypic variation is related to cladogenetic events, we performed tests for phylogenetic independence and determined the strength of convergence for color characters in relation to water type, as the variation between clear, black and white waters is considered to be one of the major driving forces in the evolution of Amazonian fishes. Color characters for fins above the median line of the body were generally found to be independent from phylogeny and the Wheatsheaf test strongly supports convergence of the dorsal fin color between populations of species in the same type of water, with a similar trend suggested for the color of the dorsal lobe of the caudal fin. This means that simple color characters cannot necessarily be relied upon for taxonomic revisions of the genus as local phenotypic variants may represent environmentally determined plasticity or convergent evolution. Further studies are required to determine the validity of these characters.

## Introduction

Amazonian waters are classically divided into blackwaters, clearwaters and white waters based on their appearance and chemistry [1], and the fish fauna of each water type is known to be generally distinct [2]. However, there is considerable variation in chemistry and the aquatic light environment both between and within these broad classifications. The role of such environmental variation in the evolution of Neotropical fishes has been revealed to be important for generating and maintaining biodiversity [3–7].

**Funding:** The authors thank CAPES (Coordenação de Aperfeiçoamento de Pessoal de Nível Superior - Brasil - Finance code 001) for studentships for ASG, SCS, ACGR, SMG and DFJS through the postgraduate programmes PPGEAP and PPGBA), and research funding from CNPq (Conselho Nacional de Desenvolvimento Científico e Tecnológico) through the Brazilian Barcode of Life (BrBOL, process 64953/2010-5) and Research Productivity Grant (process 313834/2021-0), FAPESPA (Fundação Amazônia de Amparo a Estudos e Pesquisas) Programa Primeiros Projetos grant 011/2009, FAPERGS (Fundação de Amparo à Pesquisa do Estado do Rio Grande do Sul) ARD/ ARC process 72550.751.48979, Vale (Centro de Triagem de Invertebrados contract R100603. CT.02), Hydro through the Biodiversity Research Consortium Brazil-Norway (BRC project 16/19) and National Science Foundation (NSF DEB-1146374). The funders had no role in study design, data collection and analysis, decision to publish, or preparation of the manuscript.

**Competing interests:** The authors have declared that no competing interests exist.

Many small shoaling fishes of the Neotropics present variations on a generalized color pattern with yellow/orange/red fins and silvery bodies with or without dark, melanin pigmented patches, suggesting that this common, generalized phenotype may have a role in defense against predation through disruptive camouflage and may also result in motion dazzle [8]. That pattern is largely shared by representatives of the Characiformes, among which the species of *Bryconops* Kner, 1858, are often found in mixed shoals with congenerics, and frequently with species in the *Moenkhausia lepidura* group (*sensu* [9]) and/or other species of Characidae. The genus includes Cis-andean small to medium sized tetras widely distributed in the Orinoco, Amazonas, Tocantins-Araguaia, Paraná-Paraguai, São Francisco rivers and several coastal basins draining from the Brazilian and Guiana shields [10–12]. *Bryconops* is included in the family Iguanodectidae along with *Iguanodectes* Cope, 1872 and *Piabucus* Oken, 1817 [13, 14].

Despite the recent clarification of its phylogenetic position, monophyly and interspecific relationships within the genus have never been tested satisfactorily [14–16], and its species have been traditionally assigned to two subgenera based on morphology: *Bryconops*, with short maxillae and usually lacking maxillary teeth; and *Creatochanes* Gunther, 1864, with long maxillae and usually presenting up to three maxillary teeth [17]. The subgenus *Bryconops* includes the species *Bryconops alburnoides* Kner, 1858, *B. caudomaculatus* (Günther, 1864), *Bryconops collettei* Chernoff & Machado-Allison, 2005, *Bryconops disruptus* Machado-Allison & Chernoff 1997, *Bryconops durbinae* (Eigenmann, 1908) *Bryconops gracilis* (Eigenmann, 1908), *Bryconops hexalepis* Guedes et al., 2019 [18], *Bryconops magoi* Chernoff & Machado Allison, 2005, *Bryconops piracolina* Wingert & Malabarba, 2011, *Bryconops rheorubrum* Silva-Oliveira et al., 2019, and *Bryconops tocantinensis* Guedes et al., 2016 [18]. The subgenus *Creatochanes* includes *Bryconops allisoni* Silva-Oliveira et al., 2019, *Bryconops affinis* (Günther, 1864), *Bryconops chernoffi* Silva-Oliveira et al., 2015, *Bryconops colanegra* Chernoff & Machado-Allison, 1999, *Bryconops colaroja* Chernoff & Machado-Allison, 1999, *Bryconops cyrtogaster* (Norman, 1926), *Bryconops giacopinii* (Fernández-Yépez, 1950), *Bryconops humeralis* Machado-Allison et al., 1996, *Bryconops imitator* Chernoff & Machado-Allison, 2002), *Bryconops inpai* Knöppel et al., 1968, *Bryconops melanurus* (Bloch, 1794), *Bryconops sapezal* Wingert & Malabarta, 2011, *Bryconops vibex* Machado-Allison et al., 1996 and *Bryconops marabaixo* Silva-Oliveira et al., 2019. However, various species present divergences from the putative synapomorphies defining the subgenera (e.g., *B. (B.) disruptus*, *B. (B.) piracolina*, *B. (B.) tocantinensis*, *B. (C.) inpai*, *B. (C.) marabaixo*—where the distal point of the maxilla does not reach the articulation with the quadrate), indicating a need for a revision of the genus and reevaluation of those characters in a phylogenetic framework [11, 16, 18].

In addition to morphology, identification of *Bryconops* species has been proposed to incorporate information on the pigmentation of the caudal-fin [17, 18]. However, considering the wide distribution of some taxa (especially *B. (B.) caudomaculatus* and *B. (C.) melanurus*), the potential for phenotypic plasticity associated with environmental variation [19], and the considerable array of variation associated with intensity and arrangement of melanophores within species of *Bryconops* from the middle and lower Xingu river [11], the limits between intraspecific geographic variation and species level characteristics become intrinsically difficult to distinguish.

Molecular data have been applied to many studies on Neotropical fishes (eg [13, 20–26]), and even taxonomically incomplete phylogenies can be used to verify specific taxa with identification problems, search for cryptic species and species complexes [27], assess the genetic diversity present in the study group [20, 28–31], and test phylogenetic signal and convergence of morphological traits [32, 33].

Indices that test phylogenetic signals can be divided into two groups [33]. The first group comprises autocorrelation indices without an evolutionary model including Abouheif's

*Cmean* [34] and Moran's *I* [35, 36]. They offer results that cannot be used in a quantitative interpretation while comparing different phylogenies, as the expected statistical value under the assumed model is unknown *a priori*. Even so, values closer to 1 indicate stronger relationships between trait values and phylogeny. The second group comprises indices that assume a Brownian Motion (BM) model of trait evolution including Blomberg's *K* [37] and Pagel's λ [38, 39]. For these, values closer to zero indicate phylogenetic independence, while higher values approaching or even exceeding 1 indicate that the traits are distributed in accordance with BM. Characters identified as evolving independently from the phylogeny can then be tested for coevolutionary strength using the Wheatsheaf index [40]. The Wheatsheaf index weights close phenotypic similarity highly for distantly related species by generating phenotypic distances from traits across species and then penalizes these by phylogenetic distance before investigating similarity.

The present study therefore aimed to use molecular data to test whether the phenotypic variation observed in representatives of the genus could represent overlooked diversity and be useful for taxonomic purposes, by recreating the phylogeny of this group to test if such variation is independent of the evolutionary history and is correlated to water color from the specimen's localities.

## Material and methods

*Bryconops* samples were obtained from specimens originating from four ecoregions (*sensu* [41]), corresponding to independent drainages in central and eastern Amazonia (Fig 1, S1 Table). These represent: coastal streams of the Atlantic region that show seasonal variability between dilute blackwater and clearwater classifications (ecoregion 323, eight samples); similarly variable dilute blackwater and clearwater streams of the southern Guiana shield (ecoregion 315, nine samples); more concentrated blackwater streams from the Rio Negro basin in central Amazonia, near Manaus (ecoregion 314, 25 samples); and streams of the Xingu river near Altamira that are mostly clearwater but with some turbid streams (ecoregion 322, nine samples). Preexisting samples from the clearwater Tapajós river (Pará, Brazil, two samples), the Itapecuru river (Maranhão, Brazil, 14 samples) and the São Francisco river (Minas Gerais, Brazil, three samples) were also included. Samples were collected under SISBIO collection licenses 12773–1 and 37742–1 using small seine nets and photographed using a Canon G12 camera as soon as possible after capture to record live color patterns (S1 and S2 Figs). After euthanasia with eugenol, following Lucena et al. [42] (as approved by the Federal University of Pará Animal Ethics Committee, CEUA license 682015), tissue samples (surface area of ~2mm$^2$) were removed from the right side of fish in a series of distinct positions that allow subsequent identification of individuals in mixed lots, leaving the left side intact for morphological analyses. Tissues were stored in 96% ethanol, and voucher specimens fixed in 10% formalin before transferal to 70% ethanol for long term preservation and deposition of vouchers at the Museu Paraense Emílio Goeldi (MPEG).

Identification of the material used the morphological characters defined for the genus [10–12, 17–18, 43]. Additionally, color pattern characters were defined including: the presence or absence of an ocellus on the caudal-fin upper lobe, the overall color of the caudal, adipose and dorsal fins (Yellow/Orange/Red), and the presence or absence of melanin pigmentation (Hyaline/Black). A color scale was originally included in some photos, but was not available for all specimens. To avoid possible bias in using digital coding, the color spectrum was simplified between hyaline, yellow, orange and red classification.

All new samples including representatives of *Bryconops* (N = 51) as the ingroup as well as *Iguanodectes* spp. (N = 7) and *Acestrorhynchus* sp. (N = 1) as outgroups (S1 Table) were

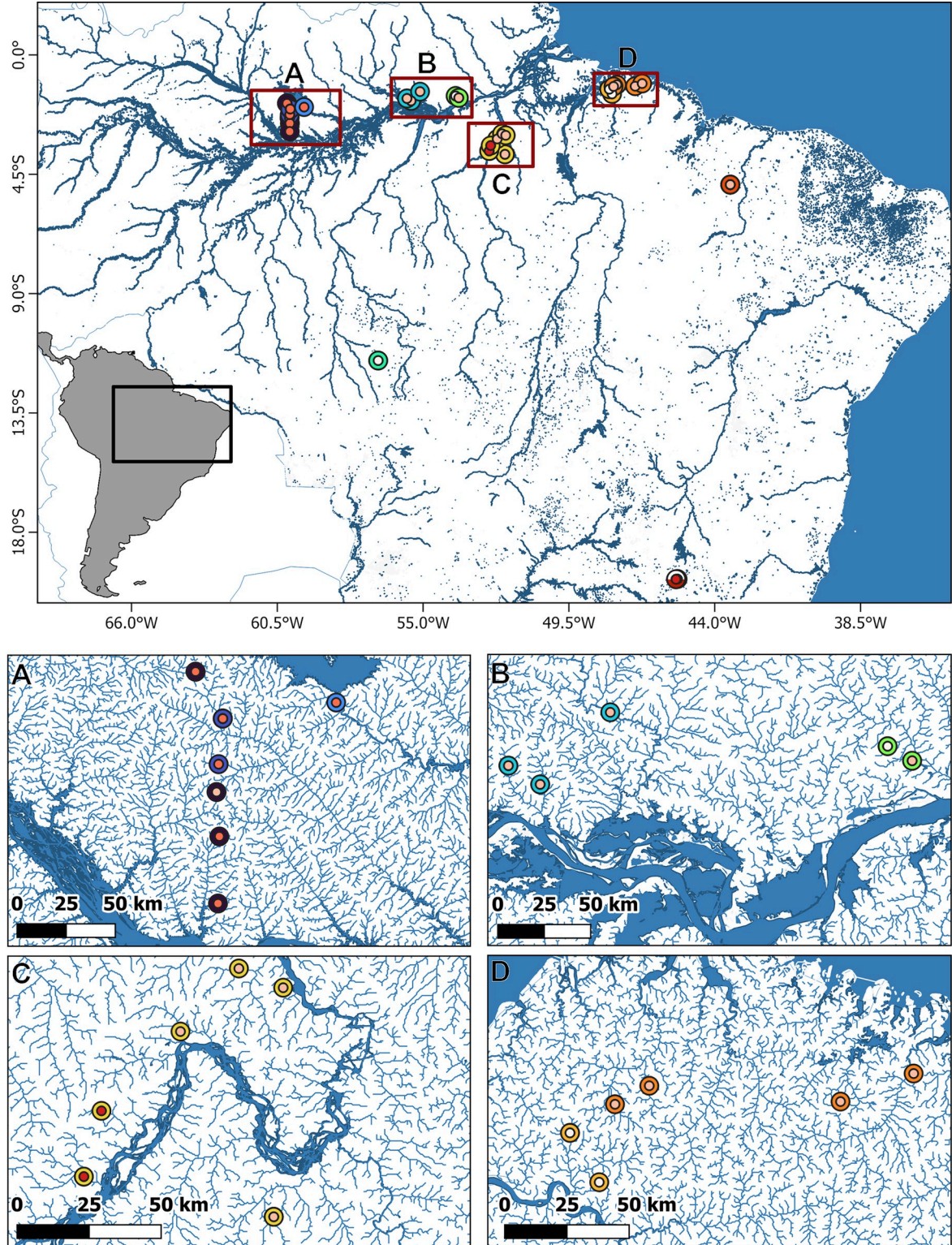

**Fig 1. Map of *Bryconops* sampling locations with indication of the water type.** Water type: Black = Turbid waters; Purple = Dark (high tannin concentration) Blackwaters; Orange = Light (low tannin concentration) Blackwaters; Cream = Clearwaters. Ecoregions [41]: A = Negro River basin; B = southern Guiana shield; C = Xingu River basin; D = coastal streams.

bidirectionally sequenced for the COI-5P barcode fragment. Genomic DNA was extracted using the Wizard Genomic DNA Purification Kit (Promega), following the manufacturers protocols. The COI-5P fragment was amplified by PCR using the primers LIICO1F 5′–GA TTTTTCTCAACTAACCAYAAAGA–3′ and LIICO1R 5′–ACTTCTGGGTGTCCGAARAAY CARAA–3′ [44] in a total volume of 12.5 μL containing 7.18 μL ultrapure water, 1.25 μL 10x Buffer, 0.75 μL MgCl2 (50 mM), 0.25 μL dNTP mix (8 mM), 0.125μl of each primer (10 μM), 0.06 μL Taq Platinum (5 u/μL, Invitrogen) and 1.0 μL of genomic DNA (10–50 ng). The PCR consisted of an initial denaturation cycle (3 min at 94°C), 40 amplification cycles (denaturation: 25 s at 94°C, annealing: 40 s at 52°C, and extension: 45 s at 72°C), and a final extension cycle (5 min at 72°C). Fragments were checked for size and band format (single vs. multiple) by electrophoresis on a 1% agarose gel before sequencing using the BigDye™ Terminator v3.1 Cycle Sequencing Ready Reaction Kit (Applied Biosystems) following standard protocols and read on the ABI 3130-Genetic Analyzer (Applied Biosystems).

We used existing COI-5P sequence data from BOLD [45] including 19 sequences identified as *Bryconops* and 6 sequences representing other members of the Iguanodectidae (*Piabucus melanostoma* N = 2) and the outgroup *Triportheus* spp. (N = 4) as a base for aligning chromatograms in Geneious v9 (http://www.geneious.com) [46], visually inspecting the alignments before production of consensus sequences for each specimen. These were then checked for stop codons and the final dataset was produced as a 661bp alignment of 84 sequences, including 70 sequences of *Bryconops* (Table 1). All new sequences were submitted to BOLD (see S1 Table for details).

The best partition parameters and evolutionary model for the dataset was determined using PartitionFinder 2 [47, 48]. Three partitions were tested, 1) one partition for all codon positions; 2) one partition for positions 1 and 2, and another for the third codon position; and 3) a partition for each codon position.

Based on these model parameters, phylogenetic trees were produced to infer the evolutionary history of *Bryconops*. A Maximum likelihood (ML) tree was made in RAxML 8.2.10 [49] using random seeds in three independent runs to avoid local topological peaks. All three generated trees showed the same topology, a bootstrap analysis was performed to verify support using 1000 bootstrap pseudo-replicates. A Bayesian Inference (BI) tree was produced using MrBayes 3.2.7 [50]. Three independent runs were performed, each with four chains and 10

**Table 1. Summary data for sample numbers by species (N Total), number of localities from which they were sampled, and water types, and number of samples by water type (N_WT).**

| Species | N Total | N localities | Water types | N_WT |
|---|---|---|---|---|
| *B. (B.) caudomaculatus* | 21 | 11 | Clearwater | 3 |
|  |  |  | Dark Blackwater | 12 |
|  |  |  | Light Blackwater | 6 |
| *B. (B.) rheorubrum* | 1 | 1 | Turbid | 1 |
| *B. (C.) affinis* | 3 | 2 | Dark Blackwater | 1 |
|  |  |  | Turbid | 2 |
| *B. (C.) aff. Affinis* | 3 | 2 | Light Blackwater | 1 |
|  |  |  | Turbid | 2 |
| *B. (C.) colaroja* | 2 | 1 | Dark Blackwater | 2 |
| *B. (C.) giacopini* | 11 | 7 | Dark Blackwater | 11 |
| *B. (C.) melanurus* | 23 | 10 | Clearwater | 4 |
|  |  |  | Light Blackwater | 19 |
| *B. (C.) sp nov 1* | 3 | 2 | Light Blackwater | 3 |
| *B. (C.) sp nov 2* | 3 | 1 | Light Blackwater | 3 |

generations with one tree sample every 1000 generations. From the total 10000 trees, we discarded the first 10% as burn-in, after checking the ESS values for all statistics were above 200 with Tracer 1.7 [51]. All analyses recovered the same topology.

To provide measures that are comparative with classical barcoding studies a standard Neighbor-Joining (NJ) tree was produced in MEGA X [52], using the Kimura 2 parameter model of evolution [53] and pairwise deletion of missing data among the samples. Support values were obtained based on 1000 bootstrap pseudo-replicates [54].

To validate the known species of *Bryconops* and to check the existence of overlooked or cryptic species, three methodologies of species delimitation were used. 1) Automatic Barcode Gap Discovery—ABGD [55], a threshold methodology to delimit species; 2) Bayesian implementation of PTP—bPTP [56], a coalescent based methodology that uses a model-based approach upon an ML or BI gene tree; and 3) General Mixed Yule Coalescent—GYMC [57–59], another model-based approach on an ultrametric gene tree.

ABGD analysis was run via the website https://bioinfo.mnhn.fr/abi/public/abgd/abgdweb.html using the default settings and Kimura (k80) model with TS/TV = 2.0. Default settings were used for the remaining parameters. bPTP was run via the website https://species.h-its.org/ptp/ using the default settings and the ML tree obtained above as input. GYMC was run via the website https://species.h-its.org/gmyc/ using the single threshold method and a bayesian ultrametric tree, without outgroups and with only one sample per haplotype, obtained from BEAST 1.10.4 [60].

Color pattern and visual environment (water type) were classified in categorical states for all samples in order to perform tests of phylogenetic independence (Abouhief's $C_{mean}$—[34]; and Pagel's $\lambda$ - [39]), as suggested by Münkemüller et al. [33] and convergence (Wheatsheaf index—[40]). For new samples, this classification was based on live color photographs taken immediately after collection. For samples for which existing sequence data was used, metadata and original photographs were requested from collectors, and if photographs were unavailable original references evaluated to use the description of color pattern associated to the voucher specimens of those sequenced samples (or from samples collected during the same collection event). The color pattern was recorded separately for melanic and non-melanic color in each fin using binomial classification (0 = no patch, 1 = patch) for Dorsal fin melanin (DF_Mel), and categorical classification (0 = Hyaline, 1 = Yellow, 2 = Orange, 3 = Red fin pigmentation) for: Color of dorsal fin (DF); Color of adipose fin (AdF); Color of dorsal lobe of caudal fin (DLCF); Color of ventral lobe of caudal fin (VLCF). The visual environment from which the samples were collected was classified based on field photographs and measurements and reported water characteristics, also following the categorical classification (0 = WT_transp, 1 = WT_few, 2 = WT_many, 3 = WT_turbid) considering four water types: WT_transp (clear/transparent waters); WT_few (waters with few dissolved tannins and a secchi disc reading of >1m); WT_many (waters with many dissolved tannins and a secchi disc reading of <1m); and WT_turbid (turbid, sediment carrying waters). All the metadata and classifications are available in S2 Table. The tests for phylogenetic independence and convergence were performed using the packages *phytools* [61], *adephylo* [62] and *windex* [63] in R [64] using the scripts "Phylogenetic significance" and "Wheatsheaf index—Phylogeny convergence" (https://github.com/TBurla/Phylogenetic-significance-and-convergence).

## Results

No indels or stop codons were identified in the new sequences or in the final aligned data matrix including publicly available data and the best partitioning scheme identified was the one where each codon position was treated independently, with the following substitution

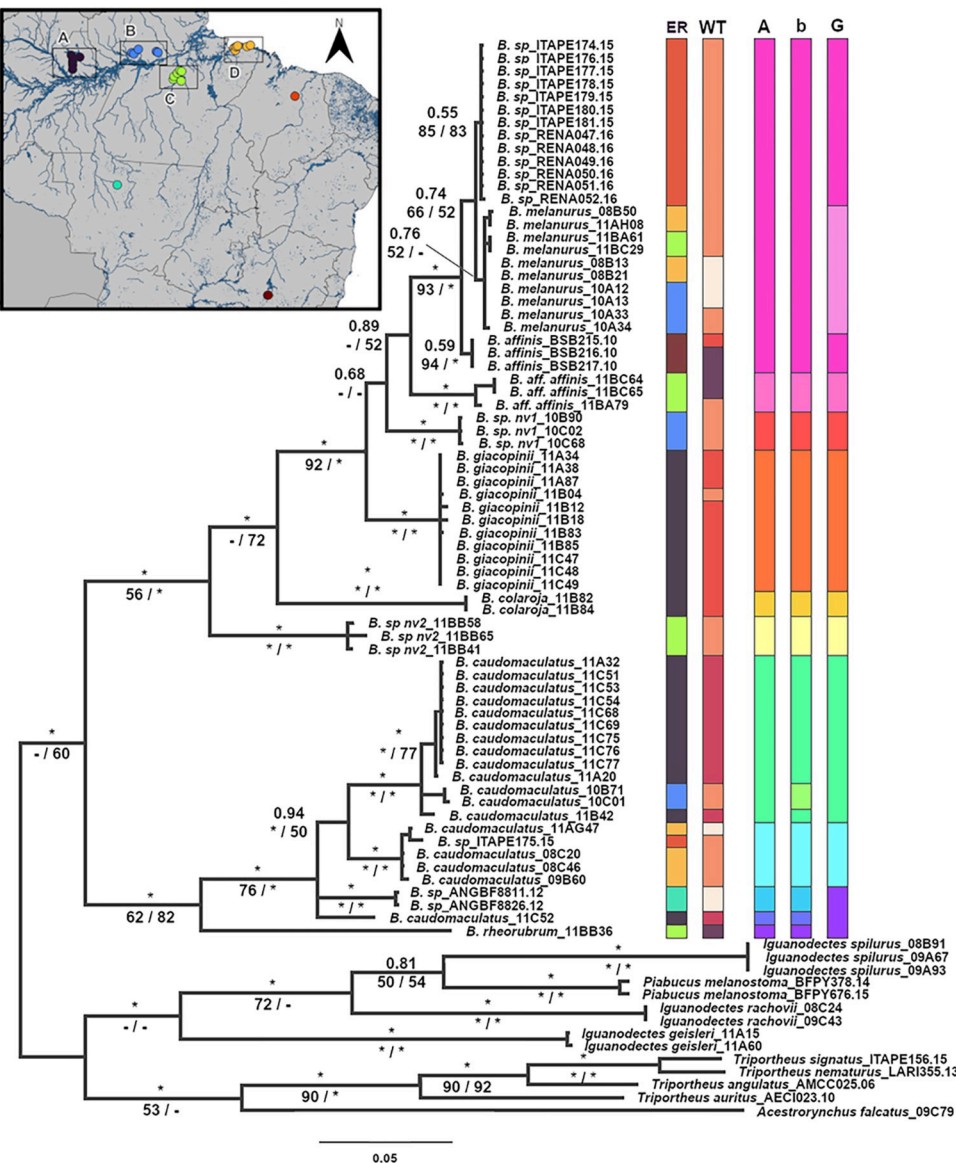

**Fig 2. *Bryconops* phylogeny.** Maximum likelihood phylogeny with vertical bars representing sample location by ecoregion (ER), water type (WT), and results of species delimitation analyses. Support values for nodes are based on Bayesian *a posteriori* probability (above) and bootstrap for ML and NJ analyses (below, separated by slash) respectively. The colors displayed for ER are in accordance with the inset map, while colors for WT are in accordance with the map from Fig 1. The Molecular Operational Taxonomic Units (MOTUs) identified by ABGD (A), bPTP (b) and GYMC (G) species delimitation methodologies were colored to highlight the subgenera *Bryconops* (purple, blue and greens) and *Creatochanes* (yellow, orange, red and pinks).* = 100% bootstrap support or BPP = 1,— = no support.

models TRN+I+G, TIM+I, and GTR+I+G, for the first, second, and third codon positions respectively under both AIC and BIC analyses.

All phylogenetic analyses (BI, ML, NJ) resulted in similar topologies, and we present the ML tree with support from the three methodologies (Fig 2). Although with low support from ML and NJ, all methodologies show the presence of the two subgenera as distinct monophyletic groups. Group one representing species in the subgenus *Bryconops* contains samples representing a species complex currently identified as *B. (B.) caudomaculatus* or *B.*

*(B.) cf. caudomaculatus* (including public sequence data), as well as *B. (B.) rheorubrum*. Group two represents the subgenus *Creatochanes* and includes samples representing the species *B. (C.) affinis*, *B. (C.) colaroja*, *B. (C.) giacopinii*, *B. (C.) melanurus* and nine specimens belonging to three unidentified species of *Bryconops* (Fig 2). One of these shows morphological similarity to *B. (C.) affinis* (*B. (C.) aff. affinis*—samples 11BA79, 11BC64 and 11BC65), whereas the other two are more distinct. Both *B. (B.) caudomaculatus* and *B. (C.) melanurus* (including public data for *B. (C.) affinis*) show substructing of lineages based on geographical origin (sample code groups, S1 Table) with deeper divergences between lineages of *B. (B.) caudomaculatus*, resulting in greater numbers of species delimited by all methods (Fig 2).

The three methodologies of species delimitation employed resulted in 10 (GYMC), 11 (ABGD) or 12 (mbPTP) MOTUs. The difference in counts between the methods result from the two widespread taxa (one extra *B. (B.) caudomaculatus* lineage under bPTP, separating samples 10B71 and 10C01, Fig 2) and one extra *B. (C.) melanurus* lineage under GMYC) and the lack of distinction of *B. (B.) rheorubrum* and basal lineages of *B. (B.) caudomaculatus* under GMYC. Also in the subgenus *Bryconops* both ABGD and bPTP show the presence of a MOTU composed of two unidentified specimens of *Bryconops* (B. sp_ANGBF8811.12 and B.sp_ANGBF8826.12), suggesting the presence of a possible new species.

For the subgenus *Creatochanes*, both phylogeny and species delimitation methodologies support the monophyly of the known species *B. (C.) giacopinii* and *B. (C.) colaroja*; there is also a concordance about the monophyly and evolutionary independence of three groups comprising three unidentified *Bryconops*, one formed by *B. (C.) sp. nv 1* (samples 10B90, 10C02 and 10C68), another formed by *B. (C.) sp. nv 2* (samples 11BB58, 11BB65 and 11BB41), and the last one by an unidentified species similar to *B. (C.) affinis*, herein named *B. (C.) aff. affinis* (samples 11BA79, 11BC64 and 11BC65) (Fig 2).

*B. (C.) melanurus* forms a monophyletic group along with public sequences for 13 unidentified *Bryconops* specimens from Maranhão state (ITAPE and RENA sample codes) and three specimens of *B. (C.) affinis* (BSB215.10, BSB216.10 and BSB217.10). While GYMC supports that *B. (C.) melanurus* should be treated as an independent species, the support values provided by the phylogenetic analyses were low for monophyly of the group including our *B. (C.) melanurus* samples and the 13 unidentified specimens from Maranhão. *B. (C.) affinis* possesses strong support as a distinct clade under both ML and NJ analyses, but no support under BI analysis and only very weak support (possible partial separation only based on GMYC) for treatment as a distinct MOTU (Fig 2).

## Are color characters independent of phylogenetic signal and, if so, do they converge in different water types?

For both the autocorrelation index and Brownian Motion model, independence from phylogenetic signal was found to be greater for the characters "Color of dorsal fin", "Color of adipose fin" and "Color of dorsal lobe of caudal fin", indicating a general trend for color pattern characters that included variation between hyaline, yellow, orange, and red color for fins above the vertical midline of the body (Table 2).

The Wheatsheaf index identified moderately strong strength of convergence for all three color pattern characters that were identified as showing independence from phylogenetic signal but with varying degrees of significance. Only "Color of dorsal fin" was found to show significant convergence with the water type from which the samples were collected, while "Color of dorsal lobe of caudal fin" approached significance (Table 2).

**Table 2. Phylogenetic independence and coevolutionary strength tests.** Test for phylogenetic independence of phenotypical traits using Autocorrelation (Abouheif's $C_{mean}$) and Brownian Motion model (Pagel's $\lambda$), and coevolutionary strength of these traits with water type as determined by Wheatsheaf test, presenting the value of the index (Wheatsheaf) with the respective lower and upper bounds and significance (*P*).

| Trait | Abouheif's $C_{mean}$ | Pagel's $\lambda$ | Wheatsheaf | Lower bound | Upper bound | *P* |
|---|---|---|---|---|---|---|
| Color of dorsal fin | 0.404 | 0.575 | 0.898 | 0.895 | 0.906 | 0.001* |
| Dorsal fin melanic patch | 0.808 | 1.000 | 0.830 | 0.827 | 0.841 | 0.854 |
| Color of adipose fin | 0.758 | 0.924 | 0.861 | 0.858 | 0.870 | 0.380 |
| Color of dorsal lobe caudal fin | 0.456 | 0.742 | 0.880 | 0.877 | 0.888 | 0.074 |
| Color of ventral lobe caudal fin | 0.669 | 1.000 | 0.836 | 0.834 | 0.847 | 0.894 |

*Significant p-value.

## Discussion

The two main clades obtained here corroborate the morphological diagnoses of the described species and subgenera [17], but clearly indicate a greater diversity in the *B. (B.) caudomaculatus* clade including at least four cryptic or overlooked species with at least partially overlapping distributions in eastern Amazonia. Much shallower, but similar divergence patterns are found in *B. (C.) melanurus* suggesting that these populations are currently exposed to isolation mechanisms that may represent the start of speciation processes. In both main clades, specimens originating from the Rio Xingu (Brazilian Shield) represent the sister groups to all other (*Bryconops*) and (*Creatochanes*), with both clades also containing lineages from all geographic regions sampled (Atlantic Coast of the Amazon, southern Guiana shield and the blackwater streams of central Amazon near Manaus). Considering the sampling limits, this follows the most common biogeographic patterns for Amazonian fishes as described by Dagosta & Pinna [21]. Moreover, the moderately large number of taxa found in each of three relatively small geographic regions (central Amazon near Manaus, Xingu River near Altamira, and the coastal rivers near Belém) suggests that dispersal capability of these taxa is high and secondary contact between species is likely to result in admixture or selective reinforcement of divergent characters. Indeed, there is an estimate of at least nine species level taxa in the genus for the lower and middle Xingu River [11].

Although the described species are monophyletic, similar looking taxa or MOTUs from different localities form paraphyletic or polyphyletic groups (e.g., *Bryconops* aff. *affinis* vs. *B. affinis* and the various MOTUs within the *B. (B.) caudomaculatus* clade). Additionally, amongst the widespread species (or species complexes) phenotypic color variants associated to geographic regions were found to exist in both major clades, with particularly striking variations observed within and among the *B. (C.) melanurus* and *B. (B.) caudomaculatus* clades where sampling density and geographic coverage were highest (S2 Fig). The phylogenetic independence tests and Wheatsheaf index analysis showed that the phylogenetically divergent lineages show true convergence in dorsal fin color as they share similar color patterns in the same water types, and especially across eastern Amazonia the co-collected samples confirm that this occurs syntopically (S1 Table). Improved phylogenetic and spatial sampling coverage may also strengthen support for convergence in the color of the other fins above the vertical midline of the body.

Convergent color patterns associated with distinct water types may represent either a selective process (selection by predators that results in convergence of syntopic prey species that gain protection through collective disruptive camouflage and/or motion dazzle or sexual selection) or environmental plasticity (pigments from food or environmental control of metabolic

processes for pigment production), or even as a combination of both mechanisms. In the first case, predation on these fishes is predicted to be dominated by larger fishes or birds, both of which are visually guided predators with many species presenting color vision involving multiple retinal cone types [65]. Therefore selected convergence associated with water type would be expected to be associated with the distinct light environment and transmission of the light reflected by these pigments in these water types. Sexual selection would normally be expected to result in distinct trends in coloration between sexes, but this was not observed in the analyzed samples of *Bryconops*.

In the case of environmental plasticity, it is important to note that pigments associated to yellows and red coloration in fishes are often derived from dietary sources of carotenoids, and that the exact hue and intensity can result from intraspecific behavioral processes such as social dominance as well as variations in diet or metabolism of carotenoids that are closely associated to environmental variation [66]. Given the generalized hue of individuals of species at each location sampled in this study (as well as during many collection trips throughout Amazonia —pers. obs. authors), the environmental effect on dietary sources or metabolism of carotenoids is the most probable cause of environmental plasticity in color patterns in this genus. It is also possible that selection acts on existing variation resulting from phenotypic plasticity [67]. To confirm this, specific experiments that control for predation, diet and light environment are needed to fully elucidate the evolutionary processes governing coloration in *Bryconops*.

The generalized utility of color pattern as a source of characters helpful to the taxonomy or identification of the species of *Bryconops* should therefore remain in question. Proposed future use of color pattern characters in these taxa should be accompanied by rigorous analyses that refute the possibility of environmental plasticity of the proposed characters. It should be noted that the character in question is the color itself and that characters based on the form of the color pattern that can be clearly described (discrete spatial delimitation of the presence or absence of color) should be more reliable. For example, *Bryconops (C.) aff. affinis* and *B. (C.) sp. nv 2* from the Xingu present clearly defined ocelli in the upper caudal-fin lobe and a more diffuse, melanic pigmentation on the lower lobe, whereas *Bryconops (C.) melanurus* and *B. (C.) sp. nv 1* presents a generalized orange/red pigmentation across the entire caudal-fin upper lobe, with no ocellus.

Amazonian rivers are known to present a diverse range of characteristics [1] and the three main water types often result in ecological clines or ecotones that are considered one of the driving forces for diversification of Amazonian fishes [68]. Understanding the role of evolution of color patterns of species within the context of water types for maintaining biodiversity is particularly important considering the human impacts such as deforestation and the construction of dams that alter both the water chemistry and visual environment.

## Supporting information

**S1 Fig. Variation in color of the *B. (Creatochanes)* clade across sample locations.** a) *Bryconops (C.) giacopinii* (Manaus—dark Blackwaters); b) *Bryconops (C.) melanurus* (Manaus— light Blackwaters); c) *Bryconops (C.) melanurus* (Coastal—light Blackwaters); d) *Bryconops (C.) aff. affinis* (Xingu—Clearwaters).
(PDF)

**S2 Fig. Variation in color of the *B. (Bryconops)* clade across sample locations.** a) *Bryconops (B.) caudomaculatus* (Manaus—dark Blackwaters); b) *Bryconops (B.) caudomaculatus* (S Guiana shield—lighter Blackwaters); c) *Bryconops (B.) caudomaculatus* (Coastal—lighter Blackwaters); d) *Bryconops (B.) rheorubrum* (Xingu—Clearwaters and turbid waters).
(PDF)

**S1 Table. Specimens details.** Table containing taxonomy information (Family, Genus and Species); curatorial information (Identifier, sampling date, collection deposited and sequence id for BOLD); and geographical information (Country, State, Eco region, Latitude and Longitude) for the new sequenced specimens, respectively.
(XLSX)

**S2 Table. Morphological characters.** Table containing information about color patter for five morphological characters and the watter type for specimen. The morphological characters Dorsal fin melanin (DF_Mel) possess two states (hyalin, and any black); Color of dorsal fin (DF), Color of adipose fin (AdF), Color of dorsal lobe of caudal fin (DLCF), and Color of ventral lobe of caudal fin (VLCF) possess four color states (hyalin, yellow, orange, and red); while Water type (WT) possess four [transparent, few tannins (secchi > 1m), many tannins or slight turbidity (secchi < 1m), and turbid].
(XLSX)

## Author Contributions

**Conceptualization:** Jonathan S. Ready.

**Data curation:** André L. Netto-Ferreira, Derlan J. F. Silva, Tibério C. T. Burlamaqui, Jonathan S. Ready.

**Formal analysis:** Andressa S. Gonçalves, André L. Netto-Ferreira, Tibério C. T. Burlamaqui, Jonathan S. Ready.

**Funding acquisition:** André L. Netto-Ferreira, João B. L. Sales, Jonathan S. Ready.

**Investigation:** Andressa S. Gonçalves, André L. Netto-Ferreira, Samantha C. Saldanha, Ana C. G. Rocha, Suellen M. Gales, Derlan J. F. Silva, Daniel C. Carvalho, João B. L. Sales, Tibério C. T. Burlamaqui, Jonathan S. Ready.

**Methodology:** Andressa S. Gonçalves, Tibério C. T. Burlamaqui.

**Project administration:** Tibério C. T. Burlamaqui, Jonathan S. Ready.

**Resources:** André L. Netto-Ferreira, Daniel C. Carvalho, João B. L. Sales, Jonathan S. Ready.

**Supervision:** André L. Netto-Ferreira, Tibério C. T. Burlamaqui, Jonathan S. Ready.

**Visualization:** Andressa S. Gonçalves, André L. Netto-Ferreira, Tibério C. T. Burlamaqui, Jonathan S. Ready.

**Writing – original draft:** Andressa S. Gonçalves, André L. Netto-Ferreira, Tibério C. T. Burlamaqui, Jonathan S. Ready.

**Writing – review & editing:** Andressa S. Gonçalves, André L. Netto-Ferreira, Samantha C. Saldanha, Ana C. G. Rocha, Suellen M. Gales, Derlan J. F. Silva, Daniel C. Carvalho, João B. L. Sales, Tibério C. T. Burlamaqui, Jonathan S. Ready.

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
