## [Decision Letter · Decision Letter 0]

9 Aug 2023

PONE-D-23-19903Convergent and environmentally associated plastic chromatic polymorphism in Bryconops Kner, 1858 (Ostariophysi: Characiformes: Iguanodectidae)PLOS ONE

Dear Dr. Burlamaqui,

Thank you for submitting your manuscript to PLOS ONE. After careful consideration, we feel that it has merit but does not fully meet PLOS ONE’s publication criteria as it currently stands. Therefore, we invite you to submit a revised version of the manuscript that addresses the points raised during the review process. Please address all the concerns raised by the reviewers. In particular, please note the reviewers concerns about taxonomic identification in this group and the use of the terms phenotypic plasticity and convergence in the abstract and introduction. Where you disagree with the reviewers, please include a detailed response to avoid further delays in the review process. 

We look forward to receiving your revised manuscript.

Kind regards,

Windsor E. Aguirre, Ph.D.

Academic Editor

PLOS ONE

“The authors thank CAPES (Coordenação de Aperfeiçoamento de Pessoal de Nível Superior - Brasil - Finance code 001) for studentships for ASG, SCS, ACGR, SMG and DFJS through the postgraduate programmes PPGEAP and PPGBA), and research funding from CNPq (Conselho Nacional de Desenvolvimento Científico e Tecnológico) through the Brazilian Barcode of Life (BrBOL, process 64953/2010-5) and Research Productivity Grant (process 313834/2021-0), FAPESPA (Fundação Amazônia de Amparo a Estudos e Pesquisas) Programa Primeiros Projetos grant 011/2009, FAPERGS (Fundação de Amparo à Pesquisa do Estado do Rio Grande do Sul) ARD/ARC process 72550.751.48979, Vale (Centro de Triagem de Invertebrados contract R100603.CT.02), Hydro through the Biodiversity Research Consortium Brazil-Norway (BRC project 16/19) and National Science Foundation (NSF DEB-1146374).

NO”

4. We note that Figures 1 and 2 in your submission contain [map/satellite] images which may be copyrighted. All PLOS content is published under the Creative Commons Attribution License (CC BY 4.0), which means that the manuscript, images, and Supporting Information files will be freely available online, and any third party is permitted to access, download, copy, distribute, and use these materials in any way, even commercially, with proper attribution. For these reasons, we cannot publish previously copyrighted maps or satellite images created using proprietary data, such as Google software (Google Maps, Street View, and Earth). For more information, see our copyright guidelines: http://journals.plos.org/plosone/s/licenses-and-copyright.

1. You may seek permission from the original copyright holder of Figures 1 and 2 to publish the content specifically under the CC BY 4.0 license. 

5. Please remove your figures from within your manuscript file, leaving only the individual TIFF/EPS image files, uploaded separately. These will be automatically included in the reviewers’ PDF.

Reviewers' comments:

Reviewer's Responses to Questions

**Comments to the Author**

1. Is the manuscript technically sound, and do the data support the conclusions?

Reviewer #1: Partly

Reviewer #2: Partly

2. Has the statistical analysis been performed appropriately and rigorously? 

Reviewer #1: N/A

Reviewer #2: Yes

3. Have the authors made all data underlying the findings in their manuscript fully available?

Reviewer #1: No

Reviewer #2: Yes

4. Is the manuscript presented in an intelligible fashion and written in standard English?

Reviewer #1: Yes

Reviewer #2: Yes

5. Review Comments to the Author

Reviewer #1: Review of Gonçalves et al. “Convergent and environmentally associated plastic chromatic polymorphism in Bryconops”

The article aims to test whether color is plastic in Bryconops species or if phenotypic variation is related to cladogenetic events. The work is well written and can be a good contribution. However, it presents a problem that needs to be corrected (see below). Therefore, my overall recommendation is either a major revision and resubmit.

The biggest problem has to do with the identity of the species and the consequences for the article's assertions.

The main hypothesis of the work is linked to the supposed wide distribution of two species (B. caudomaculatus and B. melanurus), which could present polymorphism in the coloration pattern. My comments here do not cast doubt on the hypothesis presented by the authors, but rather on how they use the taxonomy of the group under study.

1) The COI seems like a good tool to demonstrate hidden diversity. However, does COI present sufficient levels of resolution for the recovery of relationships in any of the lineages?

2) Although many authors have mentioned these two taxa as widely distributed, specialists in the group point to the opposite. Chernoff and Machado-Allison (2005), for example, mentioned B. caudomaculatus as restricted to the Essequibo River and other similar species occurring in the Amazon basin. Chernoff et al. (1994) mentions that B. melanurus occurs only in coastal drainages of the Guianas. Silva-Oliveira (2020) in a broad taxonomic review of the genus, confirms the presence of the true B. melanurus only in coastal drainages of the Guianas and recognizes two species complexes (B. caudomaculatus and B. melanurus) with still underestimated diversity. Probably the authors are working with undescribed species and still without a clear taxonomic delimitation. Although Silva-Oliveira’s work has not yet been published, it is important that Gonçalves et al. seek more up-to-date information on the taxonomy of the species they are using as a model.

3) Bryconops rheoruber from the supplementary material looks like B. magoi or B. collettei. According to Silva-Oliveira et al. (2019), Bryconops rheoruber presents dorsal fin originates anterior to a vertical through the first pelvic fin ray.

4) Images of some of the species used in the study are missing. As coloration is the focus of the work, all species should be presented.

Other comments are in MS.

Reviewer #2: Gonçalves et al present an interesting study examining whether color patterns in the iguanodectid fish genus Brycanops exhibits convergence or has a strong phylogenetic signal and thus can be used as a character for taxonomic delimitation. The authors score colors from specimens collected from different water types and use sequences of COI to create a phylogeny. Species partitioning methods, tests of phylogenetic signal, and a test of convergence are then employed to test the hypothesis. Although a more robust phylogeny using more genes or genomic methods and the inclusion of more specimens would have been good, the methods seem robust to make the point the authors advocate. The point that phenotypic similarity of ecologically important traits may often be the result of convergence and may complicate phylogenetic inference or species delimitation using these traits is also an important one to make.

I do have some points that I think should be addressed, which are presented below.

-I was disappointed that there was not a figure in the paper itself with the characters scored that showed the traits and the color variation. Two figures are presented in the supplemental files but given how central they are to understanding the results, I strongly encourage the authors to combine these into one figure and include them in the manuscript itself. Putting them in the supplemental files just makes things more difficult for readers. In addition, PLoS one will have a general readership and many may have difficulty understanding the traits without seeing them.

-The abstract and some parts of the paper are written in a way that makes it seem like convergence and plasticity are synonyms, which is very misleading. For example, from the abstract (lines 26-28): “…To test

whether color is plastic in those species or if phenotypic variation is related to cladogenetic events…“. What is being tested is whether the trait is related to the phylogenetic relationships of the species or not. However, there is no way of knowing if it is phenotypically plastic, as the authors indicate themselves in the Discussion. Phenotypic plasticity refers to non-genetic changes in the phenotype, that is, changes in gene expression patterns that result in different phenotypes in fish that are identical for the underlying alleles. Convergence generally refers to convergent evolution, the independent evolution of similar phenotypic traits, often related to adaptation to similar environmental characteristics. They are different things. The text should be reviewed for how this is phrased, especially the abstract and introduction (although there may be problems in other sections). The authors should indicate that they are testing for phylogenetic independents and conducting a test of phenotypic convergence, with the plasticity text saved for the discussion, where they indicate possible causes and that it is not possible to distinguish between phenotypic plasticity and evolutionary convergence with these data.

Lines 55-57: Oliveira et al. (2011) is listed as the latest phylogenetic hypothesis of the Characiformes, but this is a phylogeny of the Characidae and there are more recent phylogenies of Characids (e.g., Mirande 2018). This should be reworded.

-Lines 102-104: I assumed that saying that traits are phylogenetically independent is the same as saying that they are evolving according to Brownian motion. If they are phylogenetically dependent, they would be more similar than null expectations under Brownian motion, right? What am I missing? Can this be explained more carefully since the topic can be confusing.

-Line 118: Indicate how many specimens from how many sites are included. The info is in the tables but would be good to make it explicit up front and not make the reader have to count.

-Lines 144-147: How was color standardized, was a color scale included in the photographs? If not, lighting under which the pictures were taken may interfere with the ability of the authors to score the colors accurately. This should be addressed in the text.

-Table 1: Please indicate how many individuals were collected from each water type. This information can easily be included given the structure of the table and will be helpful for the reader.

-Line 200 should be ABGD, not ABDG. Please check other parts of the manuscript for this typo.

-Lines 200-202, the ABGD method has several parameters one sets. What were the other values used and why? If the other methods have options for other parameters, these should be indicated as well. Otherwise, make sure to note that the defaults were used.

-Line 211 also talks about scoring colors, so addressing how color was standardized could be done here.

-Line 262: Has MOTU been defined already? It should be defined before it is used.

-Line 295: add a comma after “Brownian Motion models”.

-Lines 355-356: What if the water color leads to similar sexual preferences because similarities of how the fish colors contrast with particular water colors? Maybe some colors patterns stand out more in some water types. That could lead to a correlation between water types and sexual preferences for particular fin color patterns.

6. PLOS authors have the option to publish the peer review history of their article (what does this mean?). If published, this will include your full peer review and any attached files.

Reviewer #1: No

Reviewer #2: No

---

## [Author Response · Author response to Decision Letter 0]

7 Oct 2023

We accepted several changes proposed by the revisor, we tried to remove ambiguities focusing the scope of the work in the tests made with an interesting group of study for it. Some of the reviewers arguments were replied in the "Response to Reviewers" doc, and we hope the responses can satiate theirs doubts and inquiries.

We put here the answers for the principal issues presented by both reviewers.

Reviewer 1

1- COI data has previously been shown to provide a reasonable level of information for resolving recent divergence in many fish groups from South America. furthermore, as the analysis depends on the strength of the formation of clades and this is determined by the support of nodes within the phylogeny (which in turn depends on the information available in any given dataset) we can consider that the dataset is informative as there is a clear separation of clades with generally good node support and consistent with the deeper level separation of subgenera as described in morphological studies.

2- Whether or not the species are described does not influence the interpretation of convergence (though does impact discussions of plasticity - see point 1 of reviewer 2). a fully resolved taxonomy of all samples is an unrealistic target for inclusion in this work in the near future as there are many unsampled regions and classifications are always based on current available published information and valid species. We use the taxonomic classifiers aff and cf to recognize discrepancies and variants where possible considering both the morphological identifications made and molecular lineages identified.

3- The identification was made according to the original description of B. rheoruber in Silva-Oliveira et al. 2019 (doi.org/10.1111/jfb.14445).

4- Unfortunately we don’t have all the photos within Belém Group, although we asked them for the others authors, we don’t have answers. The photos in Figures S1 and S2 represents all the species we have pictures (we have more pictures of the same species that we judge would only pollute and make difficult to understand if we put all of them together).

Reviewer 2

1- We altered the text to exclude the presence of phenotipic plasticity and convergence from abstract and introduction, putting them only in discussion.

2- saying that traits are phylogenetically independent is the same as saying that they are evolving according to Brownian motion? - Assuming a trait as a phylogenetically independent is the same as saying that it is not evolving in accord to neutral assumption, not evolving accord to Brownian Motion implied by some of the tests. If so, there is a force (with strength, theoretically, proportional to the variation in the indices) influencing the phenotype, this could be a result from sexual selection (as you suggested), from difference in predation due to mimetism, from dietary origin, or a combination of those factors. In this work we report that such possibility exists, but we would need more samples from the same localities to test if there is a deviation from Hardy-Weinberg equilibrium, considering we have the right genetic markers for it.

3- A color scale (Spyder Checkr) was originally included in some but not all photographs and was not available for some collections. Given our experience that the variation observable was not quantifiable directly using digital color coding but that within the samples we did have color standards the effects of light intensity/shade did not alter a simple classification between yellow, orange, and red coloration, we opted to continue the analysis considering this gross classification that could be reliably separated and tested.

---

## [Editor Report · Decision Letter 1]

15 Nov 2023

PONE-D-23-19903R1Convergent and environmentally associated plastic chromatic polymorphism in Bryconops Kner, 1858 (Ostariophysi: Characiformes: Iguanodectidae)PLOS ONE

Dear Dr. Burlamaqui,

Thank you for submitting your manuscript to PLOS ONE. After careful consideration, we feel that it has merit but does not fully meet PLOS ONE’s publication criteria as it currently stands. Therefore, we invite you to submit a revised version of the manuscript that addresses the points raised during the review process.

 Note the comments below. One thing that needs special attention is why some statistics and P values were removed from Table 2. The results seem to have changed between the original version and the revision. This needs to be explained very carefully since it gets at the analysis that relates to the most important points made.  Please submit your revised manuscript by Dec 30 2023 11:59PM. If you will need more time than this to complete your revisions, please reply to this message or contact the journal office at plosone@plos.org. Please include the following items when submitting your revised manuscript:A rebuttal letter that responds to each point raised by the academic editor and reviewer(s). You should upload this letter as a separate file labeled 'Response to Reviewers'.A marked-up copy of your manuscript that highlights changes made to the original version. You should upload this as a separate file labeled 'Revised Manuscript with Track Changes'.An unmarked version of your revised paper without tracked changes. You should upload this as a separate file labeled 'Manuscript'.

We look forward to receiving your revised manuscript.

Kind regards,

Windsor E. Aguirre, Ph.D.

Academic Editor

PLOS ONE

Additional Editor Comments:

The page and line numbers refer to the clean version of the manuscript in revision 1.

-For the title: I do not think that you have shown whether the pattern is the result of plasticity (non-genetic phenotypic change), so it seems misleading to include plasticity in the title. I think you can just remove the plastic from the title and leave it as: “Convergent and environmentally associated chromatic polymorphism in Bryconops Kner, 1858 (Ostariophysi: Characiformes: Iguanodectidae)”. The “environmentally associated” leaves the possibility of plasticity without actually using the word without showing that it is plasticity.

-Line 27: Add a comma “,” after cladogenetic events.

-Lines 34-37: For the last line of the abstract, it could be plasticity or convergent evolution. Either could cause problems for grouping or identifying species. I think you need to add the possibility of convergent evolution to the sentence and suggest modifying it to:

“This means that simple color characters cannot necessarily be relied upon for taxonomic revisions of the genus as local phenotypic variants may represent environmentally determined plasticity or convergent evolution. Further studies are required to determine the validity of these characters.”

-Lines 53-57: The modification to include “actually” does not work very well the way the word is used here. Although it is a long sentence, I suggest changing to:

“The genus includes Cis-andean small to medium sized tetras widely distributed in the Orinoco, Amazonas, Tocantins-Araguaia, Paraná-Paraguai, São Francisco rivers and several coastal basins draining from the Brazilian and Guiana shields [10-12]. Bryconops is currently included in the family Iguanodectidae along with Iguanodectes Cope, 1872 and Piabucus Oken, 1817 [13-14].”

-Line 80: Replace “demand” with “need”.

-Lines 112-113: Change to: “to test if such variation is independent of the evolutionary history and is correlated with water color from the specimen’s localities.”

-Line 148: Divide into two sentences. So: “photos, but was not available for all specimens. To avoid possible bias in using digital”

-Table 1 as currently formatted is confusing because the records are not clearly separated between species. Please make the separation between species easier to see.

-Line 203: Modify to: “(k80) model with TS/TV = 2.0. Default settings were used for the remaining parameters. bPTP was run via”.

-Line 211: Add a comma “,” after “For new samples,”

-Table 2: I assume that the lower and upper bounds and P are for the Wheatsheaf test. Please indicate so in the caption o there is no doubt. Also, not that for the Dorsal fin melanic patch, the numbers are on two different lines. The same happens for the two last traits. I assume this is not intentional and should be corrected.

-Table 2: Why were the P values removed from Abouheif’s C mean and Pagel’s test? Why were Moran’s I and Blomberg K removed? In the previous versions, these tests indicated strong significance. Now there is no indication of this. The actual numerical values have also changed. In the new version, only the color of the dorsal fin has a P value of less than 0.05. This needs to be explained carefully to ensure that the results still hold true since this involves the key component of the analysis and interpretation of convergence in these color traits.

-I would have liked to see figures S1 and S2 merged into one figure and included in the main manuscript. I think the pictures are beautiful and help the non-specialist understand the color characters better. However, I will not insist if the authors prefer not to make this modification for some reason. Just make sure to update the figure numbers if you do.

---

## [Author Response · Author response to Decision Letter 1]

29 Dec 2023

We changed the title as requested.

About the text, we appreciate the suggestions and changed following them. We also review the text and add some minor changes, all presents in the Response to Reviewers.

Lines 124-128 in the current revision. We improved the text to clarify the new sampling targeted for this study and the preexisting samples that were possible to include because of available and informative metadata.

We tried to make it easer to read the Table 1, formatting the table with ditto marks to differentiate the species and changed the order of locations column.

About Table 2, we discussed in Response to Reviewers, but in brief:

The changes in the numbers present between the former and actual Table 2 for Abouheif’s and Pagel’s tests were generated because the data presented in the previous table was generated with part of the total data while we were testing our code. Once we finished it, we produce the results, with the complete dataset, present in the actual Table 2. We removed the P values for both tests because the most important thing are how close those results are in relation to 0 (independence from phylogeny) and 1 (evolving in accord to phylogeny) are. The significance is only important for the correlation test, which we keep.

About the removal of Moran’s and Blomberg testes, we acted in accord to Münkemüller et al. (2012 - https://doi.org/10.1111/j.2041-210X.2012.00196.x) keeping the more stable and with greater overall performance.

If compared to the previous Table 2, it will be seen that the trend didn't changed for any trait observed. And if compared to the previous Table 3, it will be seen that the only correlation test with significance is related to 'Color of dorsal fin'. So, even though we merged the two tables, the results are comparable, but in the actual table 2 they come from all of the data, and not only a part as in the previous.

---

## [Editor Report · Decision Letter 2]

22 Jan 2024

Convergent and environmentally associated chromatic polymorphism in Bryconops Kner, 1858 (Ostariophysi: Characiformes: Iguanodectidae)

PONE-D-23-19903R2

Dear Dr. Burlamaqui,

We’re pleased to inform you that your manuscript has been judged scientifically suitable for publication and will be formally accepted for publication once it meets all outstanding technical requirements.

Kind regards,

Windsor E. Aguirre, Ph.D.

Academic Editor

PLOS ONE

Additional Editor Comments (optional):

Congratulations on the acceptance of the article. Please work with the editorial office to correct two minor details in the abstract:

Line 33: Should be "...with a similar trend"...". The "a" is missing in the abstract text.

Line 36: Remove "rather than local selection". This is not needed to understand the meaning of the text and local selection can lead to convergent evolution, so these are not really alternatives.
---

## [Editor Report · Acceptance letter]

6 Feb 2024

PONE-D-23-19903R2 

PLOS ONE

Dear Dr. Burlamaqui, 

I'm pleased to inform you that your manuscript has been deemed suitable for publication in PLOS ONE. Congratulations! Your manuscript is now being handed over to our production team.

Kind regards, 

on behalf of

Dr. Windsor E. Aguirre 

Academic Editor

PLOS ONE